# REDUCED-RANK ONLINE GAUSSIAN PROCESS MODELING WITH UNCERTAIN INPUTS

## ABSTRACT

Gaussian Process (GP) is an increasingly popular modeling approach. In its classical formulation, the inputs are supposed to be perfectly known. However, in some use cases, this assumption is not true: the inputs as well as the outputs can be corrupted by noise. Some methods already insert these uncertainties in GP modeling but the only currently existing online algorithm (i.e. that incrementally updates the model each time a measurement is acquired) still lacks in robustness and precision. In this article we propose a novel online Gaussian Process (GP) modeling approach for vector field mapping with uncertain inputs. They are included into the GP through a complete second-order Taylor approximation with a better estimation of variances. Our experiments prove that our algorithm is more accurate and robust than the previous online method for a shorter computing time. Moreover, for high input uncertainties, our method achieves better performance than both online and offline state of the art methods on simulated data. This algorithm can also be applied to diverse real scenarios which require precise estimation of unknown functions from a small set of corrupted datapoints, as we show in the challenging problem of indoor localization, mapping magnetic fields.

## 1 INTRODUCTION

The problem of data interpolation is crucial for accurate modeling of a field from discrete data points. Over the years, Gaussian processes (GP) have become an increasingly popular interpolation method due to their flexibility and their inherent ability to give a reliable estimate of their own uncertainty. However, the classical GP modeling is based on several assumptions: the values of the inputs are perfectly known and only the outputs are corrupted by a Gaussian white noise, identical for all the different datapoints (Rasmussen & Williams, 2006; Quiñonero-Candela & Rasmussen, 2005). In this paper we will focus on cases where the first hypothesis is not verified, which is a common issue in real-life applications of GPs: for example, satellite data is often associated to a particular position on Earth but the images' resolution can be low and therefore a source of uncertainty that should be taken into consideration (Cressie & Cornac, 2003; Arbia et al., 1998). Modeling maps from sensors for robotics applications, an important task in engineering, has a similar issue (Kersting et al., 2007). In Simultaneous Localization And Mapping (SLAM) problems, the map creation is based on uncertain inputs (Mourikis & Roumeliotis, 2004; Kok & Solin, 2018).

To address this issue, different models (eg. Mchutchon & Rasmussen (2011); Cressie & Cornac (2003); Bijl et al. (2017)) have been proposed to incorporate the uncertainty of the inputs. However, each has benefits and limitations in terms of performance, robustness and computational resources, three crucial properties for practical applications. Indeed, non-online methods require the inversion of large matrices leading to a high computational complexity. On the other hand, the existent online method (Bijl et al., 2017) lacks in robustness due to numerical instabilities: for high input uncertainties, we observe the appearance of very high gradients in the estimated functions which are not realistic and the algorithm diverges. This will be further discussed in the sub-section 5.1.

**Contributions.** In numerous applications, real-time estimation of a model are needed (for example patients' vital signs, localization or telemetry data). Therefore a GP model that can be used online, with updates of a short computing time for each new measurement, and with a strong reliability (i.e. with a low risk of divergence) is required. The previously existing methods including uncertain inputs did not match these requirements and we created a new GP algorithm for uncertain inputs that

includes ideas from a special category of GP modeling, never studied in this context: a method that uses smooth eigenfunctions in order to avoid divergences that could rise. We confirm the relevance of this approach by a comparison on simulated data between our algorithm and other noisy inputs GPs. It showed a better robustness and speed compared to the other existent online algorithm while also having a better accuracy. We hope that this opens the way to study new characteristics of reduced-rank GP methods. Moreover, to assert the practical use of this algorithm, we also adapted and evaluated it on a challenging real-life problem, a very active field in robotics: online localization in indoor environments, and obtained better results than the state of the art Coulin et al. (2022). Moreover, to assert the practical use of this algorithm, we also adapted and evaluated it on a challenging real-life problem: online localization in indoor environments, and obtained better results than the state of the art.

This paper is structured as follows. Section 2 is an overview of the existing GP algorithms including uncertain inputs. In the Section 3, the fundamental hypotheses for our approach with uncertain inputs are detailed, while in Section 4 our reduced-rank GP algorithm is detailed, as well as its adaptation for the particular use case of online magnetic map building. We finally evaluate our method in Section 5.

## 2 RELATED WORK

In this section we present the different approaches that are used for GP estimation in the presence of both input and output noises on the data.

**Modifying ouput noise.** Some approaches focused on the inhomogeneity of the output noise: they estimate the sensor noise according to the data collected, assuming that this is one random variable more to be added in the model (Goldberg et al., 2011; Kersting et al., 2007). These methods could be applied to describe data subject to input noise but nonetheless overfitting effects are difficult to avoid and it does not enable to include directly the knowledge of the input uncertainty.

**Input noise modeling.** Several methods were proposed to modify the GP modeling in order to include the input noise and demonstrated that the estimations lose in precision when the input noise is not integrated in the equations (Cressie & Cornac, 2003; Mchutchon & Rasmussen, 2011). In Cressie & Cornac (2003); Cervone & Pillai (2015); Dallaire et al. (2009), a Gaussian white noise is added to the inputs and the covariances are modified accordingly. The expected value of the process with position uncertainty is modeled as the integral of the expected value of the process without uncertainty over all the possible locations, weighted by the probability density of the location error. The main drawback of this method is that most of the time, these integrals cannot be solved analytically and for numerical estimation, a Monte-Carlo density estimation is proposed. This method has a high accuracy but requires a large number of covariance computations. Other approaches are less costly and use Taylor series decomposition (Mchutchon & Rasmussen, 2011; Bijl et al., 2017): the estimations are significantly better than a classical GP. However the matrices obtained from the models mentioned above grow as the number of data increases. It becomes quickly computationally intractable.

**Online approach.** Therefore an algorithm based on sparse Gaussian processes (Gijsberts & Metta, 2013; Bui et al., 2017), adapted to online estimation, is proposed in Bijl et al. (2017): the idea of sparse GPs is to compute and store the mean and the covariance at some particular points, called nodes. These nodes are generally uniformly scattered on the function domain. The studied function can be estimated everywhere interpolating the results at the stored nodes (Bijl et al., 2015). This method is an approximation of the complete GP: its predictions are slightly less accurate but its computing time is greatly reduced and much more tractable for high number of datapoints. Although it is still better than the previously existing algorithms, the model is too confident in its predictions and has a tendency to diverge often when the input noise is not small and this phenomenon is amplified when a lot of datapoints are used for learning. This is why we propose here another algorithm, more robust, which is based on smooth eigenfunctions to interpolate the true function. Moreover, our method is more accurate, using a more complete version of the Taylor approximation, and does not require a first step of reestimation of the input variance, as was the case in Bijl et al. (2017).

**Application to online localization.** The problem of online localization in indoor environments is a very active research area (Arévalo, 2018; Campos et al., 2021; Coulin et al., 2022). Recent approaches use a magnetometer, coupled with other sensors, for indoor localization. Maps of the magnetic field are obtained through classical online GP estimation from pairs of position and

associated magnetic value. Nonetheless, the positions are estimated by a filter and not perfectly known, resulting in contradictions in the data and important errors in the modeling. Until now, this problem was smoothed by artificially increasing the sensor noise for the model but this method is not reliable when we do not already have an intuition of the amplitude of the position errors given by the system. Therefore a method that includes uncertain inputs and robust for high amount of data (as the space covered is often a whole building) is crucial to achieve better localization accuracy.

## 3 GAUSSIAN PROCESS WITH UNCERTAINTY ON THE INPUTS: PRELIMINARIES

### 3.1 OBJECTIVES

The objective is to evaluate as precisely as possible the values of a field $B$ in any point of space $\mathbf{x} \in \mathbb{R}^d$, with $d$ the space dimension. The field is supposed to be constant over time. Its dimension is equal to $c$: $B(\mathbf{x}) \in \mathbb{R}^c$. To estimate $B$, one may model it as a random vector which probabilistic distribution can be determined through different measures of the field.

In the following, the distribution of $B$ is studied using the Gaussian process theory: $B$ is described as a Gaussian process entirely defined by its expected value $m(\mathbf{x}) = \mathbb{E}[B(\mathbf{x})]$ and its covariance $\kappa(\mathbf{x}, \mathbf{x}') = \text{Cov}(B(\mathbf{x}), B(\mathbf{x}')) = \mathbb{E}\left[(B(\mathbf{x}) - m(\mathbf{x}))(B(\mathbf{x}') - m(\mathbf{x}'))^\top\right]$. In the initial stage, $m(\mathbf{x})$ is set to $\mathbf{0}_c$ because we have no a priori information on its value.

We perform a series of measurements, each one indexed by its number. The $i$-th measurement takes place in a point of space $\mathbf{x}^i$ and is in the form of a vector $\mathbf{b}^i \in \mathbb{R}^c$. Ideally, $\mathbf{b}^i$ should be equal to $B(\mathbf{x}^i)$, which would be valid only for a perfect sensor. However, the sensor's measurements are subject to a noise that we model as the realization of a Gaussian isotropic random variable, $\varepsilon$, with a variance equal to $\Sigma_b$. The measurement can be written as

$$\mathbf{b}^i = B(\mathbf{x}^i) + \varepsilon. \tag{1}$$

This equation corresponds to perfect inputs but in practice an incertitude on the input $\mathbf{x}^i$ for each $i$ should be included. The input error can vary over time. We model it by $\xi^i \in \mathbb{R}^d$ a random vector of size $d$ and of variance $\Sigma^i$, different for each measure. The measured input value is

$$\hat{\mathbf{x}}^i = \mathbf{x}^i + \xi^i. \tag{2}$$

So the measure, including input uncertainty, is modeled as

$$\hat{\mathbf{b}}^i = B(\hat{\mathbf{x}}^i) + \varepsilon = B(\mathbf{x}^i + \xi^i) + \varepsilon. \tag{3}$$

For simplicity, we drop the measurement index when not necessary for the equations' comprehension. The input noises of two different measures are considered independent.

$B_{IE}(\mathbf{x}) = B(\mathbf{x} + \xi) + \varepsilon = \hat{\mathbf{b}}(\mathbf{x})$ is the random variable associated to the measures with Input Errors (IE). The measured value $\mathbf{b}(\mathbf{x})$ is a realisation of the associated random variable $B_{IE}(\mathbf{x})$.

### 3.2 REDUCED-RANK FORMULATION FOR INITIAL SETTING

The random variable $B_{IE} = B(\mathbf{x} + \xi) + \varepsilon$ is the composition of two Gaussian processes. In order to study the distribution of $B$, we approximate $B_{IE}$ as a Gaussian process: we need to evaluate its mean and variance for any value of $\mathbf{x}$ in the region of interest. The covariance function of $B$ is supposed exponential-quadratic, a very common assumption for GP interpolation (Rasmussen & Williams (2006)). $\kappa(\mathbf{x}^\star, \mathbf{x}) = \sigma_{SE}^2 \exp\left(\frac{\|\mathbf{x}^\star - \mathbf{x}\|^2}{2l_{SE}^2}\right)$ with the two hyperparameters $\sigma_{SE}$ and $l_{SE}$. $\sigma_{SE}$ represents the maximal magnitude of the covariance and $l_{SE}$ the characteristic lengthscale.

The formulation of exact GPs has a high complexity, as mentioned in Section 1. Therefore we use truncated series of the covariance function to have a more tractable algorithm (Solin & Särkkä, 2020). For one-dimensional outputs the initial covariance with any point $\mathbf{x}$ is modeled as:

$$\kappa(\mathbf{x}^\star, \mathbf{x}) = \phi(\mathbf{x}^\star)\Lambda\phi(\mathbf{x})^\top \tag{4}$$

with $\phi(\mathbf{x}) = \left(\mathbf{x}^\top, \phi_1(\mathbf{x}), ..., \phi_q(\mathbf{x})\right)$ with the eigenfunctions $\phi_i$ obtained from the initial modeling chosen and $q$ their total number. The expression of the initial $\Lambda$ is $diag(S_{SE}(\lambda_1), ..., S_{SE}(\lambda_M))$ where

$S_{SE}$ is the spectral density of the covariance. In practice, we have to limit the study of the function to a closed domain, for example rectangular as the resulting expressions of $\phi_j$ are straightforward:

$$\phi_j(\mathbf{x}) = \prod_{k=1}^{d} \frac{1}{\sqrt{L_k}} \sin\left(\frac{\pi n_{j,k}(\mathbf{x}_k + L_k)}{2L_k}\right) \quad \text{and} \quad \lambda_j^2 = \sum_{k=1}^{d} \left(\frac{\pi n_{j,k}}{2L_k}\right). \tag{5}$$

with $L_d$ the dimension of the domain definition along the axis $d$. $n_{j,k}$ is an index set of the permutation of integers $1,\dots,m$ (Solin & Särkkä, 2020). For an exponential-quadratic covariance,

$$S_{SE}(\omega) = \sigma_{SE}^2 (2\pi l_{SE}^2)^{d/2} \exp\left(-\frac{\omega^2 l_{SE}^2}{2}\right). \tag{6}$$

The distribution of $B_{IE}$ conditioned to a series of measurements $B_{IE}(\mathbf{x}) = \mathbf{b}(\mathbf{x})$ is estimated in the next section.

## 4 REDUCED-RANK GP INCLUDING UNCERTAIN INPUTS

### 4.1 REDUCED-RANK GP FROM TAYLOR SECOND ORDER POLYNOMIAL

The joint probability distribution of $B$ and $B_{IE}$ is, for any pair of points $(\mathbf{x}^\star, \mathbf{x}^\diamond)$ and with $\mathbf{x}^\alpha$ a training datapoint:

$$\begin{bmatrix} B(\mathbf{x}^\star) \\ B(\mathbf{x}^\diamond) \\ B_{IE}(\mathbf{x}^\alpha) \end{bmatrix} \sim \mathcal{N}\left(\begin{bmatrix} \mathbb{E}[B(\mathbf{x}^\star)] \\ \mathbb{E}[B(\mathbf{x}^\diamond)] \\ \mathbb{E}[B_{IE}(\mathbf{x}^\alpha)] \end{bmatrix}, \begin{bmatrix} K_1 & K_2 \\ K_2^\top & K_3 \end{bmatrix}\right) \tag{7}$$

$$\text{with} \quad K_1 = \text{Var}\left(\begin{bmatrix} B(\mathbf{x}^\star) \\ B(\mathbf{x}^\diamond) \end{bmatrix}\right), \quad K_2 = \begin{bmatrix} \text{Cov}\left(B(\mathbf{x}^\star), B_{IE}(\mathbf{x}^\alpha)\right) \\ \text{Cov}\left(B(\mathbf{x}^\diamond), B_{IE}(\mathbf{x}^\alpha)\right) \end{bmatrix} \quad \text{and} \quad K_3 = [\text{Var}(B_{IE}(\mathbf{x}^\alpha))]. \tag{8}$$

For a measurement $\hat{\mathbf{b}}(\mathbf{x}^\alpha)$ obtained at $\mathbf{x}^\alpha$, with the uncertainty on the input, the probability distribution of $B$ for any points is modified as follows (Rasmussen & Williams, 2006).

$$\begin{bmatrix} B(\mathbf{x}^\star) \\ B(\mathbf{x}^\diamond) \end{bmatrix} \sim \mathcal{N}\left(\begin{bmatrix} \mathbb{E}[B(\mathbf{x}^\star)] \\ \mathbb{E}[B(\mathbf{x}^\diamond)] \end{bmatrix} + K_2 K_3^{-1} \begin{bmatrix} (\hat{\mathbf{b}}(\mathbf{x}^\alpha) - \mathbb{E}[B(\mathbf{x}^\star)]) \\ (\hat{\mathbf{b}}(\mathbf{x}^\alpha) - \mathbb{E}[B(\mathbf{x}^\diamond)]) \end{bmatrix}, \begin{bmatrix} \text{Var}B(\mathbf{x}^\star) \\ \text{Var}B(\mathbf{x}^\diamond) \end{bmatrix} - K_2 K_3^{-1} K_2^\top\right) \tag{9}$$

Therefore we are able to estimate the probability distribution of $B$ if we know accurately the expressions of $K_1$, $K_2$ and $K_3$, which means studying the distribution of $B_{IE}$ as well as its covariance with $B$. The second order Taylor polynomial is one convenient way to approximate $B_{IE}(\mathbf{x})$ as the first order terms are null (see Appendix).

$$B_{IE}(\mathbf{x}) = B(\mathbf{x} + \xi) + \varepsilon \simeq B + \nabla B\xi + \frac{1}{2}h(B(\mathbf{x}), \xi) + \varepsilon. \tag{10}$$

For multidimensional inputs, $h(B(\mathbf{x}), \xi)$ is a vector of size $d$ with a double sum, with $\mathbf{x}_j$ the $j$-th component of the vector $\mathbf{x}$:

$$h(B(\mathbf{x}), \xi) = \sum_{j=1}^{d} \sum_{k=1}^{d} \xi_j \xi_k \frac{\partial^2 B(\mathbf{x})}{\partial \mathbf{x}_j \partial \mathbf{x}_k}. \tag{11}$$

For simplicity we note $\mathbf{x}^\alpha = \mathbf{x}$, $\Sigma$ is the variance of $\xi$ for the measurement $\mathbf{x}^\alpha$, $\phi = \phi(\mathbf{x}) = \phi(\mathbf{x}^\alpha)$, $\phi_\star = \phi(\mathbf{x}^\star)$ et $\phi_\diamond = \phi(\mathbf{x}_\diamond)$. To begin with the expression for $K_2$

$$\text{Cov}\left(B(\mathbf{x}^\star), B_{IE}(\mathbf{x})\right) = \text{Cov}\left(B(\mathbf{x}^\star), B(\mathbf{x} + \xi)\right) \simeq \text{Cov}\left(B(\mathbf{x}^\star), B(\mathbf{x}) + \nabla B(\mathbf{x})\xi + \frac{1}{2}h(B(\mathbf{x}), \xi)\right). \tag{12}$$

Using the reduced-rank formulation and because of the independence between the different components of $\Sigma$ and $B$ and its derivatives (for detailed explanations for both $K_2$ and $K_3$, see the Appendix), we can write the explicit expressions of the matrix studied:

$$\text{Cov}\left(B(\mathbf{x}^\star), B_{IE}(\mathbf{x})\right) = \text{Cov}\left(B(\mathbf{x}^\star), B(\mathbf{x})\right) + \frac{1}{2}\phi_\star \Lambda \sum_{j=1}^{d} \sum_{k=1}^{d} \Sigma_{j,k} \frac{\partial^2 \phi^\top}{\partial \mathbf{x}_j \partial \mathbf{x}_k} \tag{13}$$

$$= \phi_\star \Lambda \phi^\top + \phi_\star \Lambda \zeta^\top \quad \text{with} \quad \zeta = \sum_{j=1}^{d} \sum_{k=1}^{d} \Sigma_{j,k} \frac{\partial^2 \phi}{\partial \mathbf{x}_j \partial \mathbf{x}_k}. \tag{14}$$

$\zeta$ is of length $q$ (for a unique point $\mathbf{x}$). For $K_3$,

$$\text{Cov}\Big(B_{IE}(\mathbf{x}^v) + \varepsilon_v, B_{IE}(\mathbf{x}^\alpha) + \varepsilon_\alpha\Big) = \text{Cov}\Big(B(\mathbf{x}^v), B(\mathbf{x}^\alpha)\Big) + \frac{1}{2}\phi(\mathbf{x}^v)\Lambda\zeta^\top(\mathbf{x}^\alpha) \tag{15}$$

$$+ \frac{1}{2}\zeta(\mathbf{x}^v)\Lambda\phi^\top(\mathbf{x}^\alpha) + \delta_{\alpha,v}(D_{\text{Var}} + \Sigma_b). \tag{16}$$

with $\delta_{\alpha,v} = 1$ if $\alpha = v$ and $\delta_{\alpha,v} = 0$ otherwise, as the noises $\xi_\alpha$, $\xi_v$ and the output noises are considered independent if $\alpha \neq v$. And we note $D_{\text{Var}} = \text{Var}\big(\nabla B(\mathbf{x})\xi\big)$.

For a conditioning with several values, $\mathbf{x}^1, ..., \mathbf{x}^N$, we define $\Phi = \begin{bmatrix} \phi(\mathbf{x}^1) & ... & \phi(\mathbf{x}^N) \end{bmatrix}^\top$ (and for $\phi_\star$ as well), we can generalize the equation above. $\Theta$ is the matrix that contains the expression of $\zeta$ for $\mathbf{x}^1, ..., \mathbf{x}^N$ with $\Theta = \begin{bmatrix} \zeta(\mathbf{x}^1) & ... & \zeta(\mathbf{x}^n) \end{bmatrix}^\top$.

When $D_{\text{Var}} + \frac{1}{2}\big(\Phi\Lambda\Theta^\top + \Theta\Lambda\Phi^\top\big)$ is not positive-definite, the measurement is ignored because there is a high risk of negative covariances: the Taylor approximation should be considered as too coarse.

## 4.2 ITERATIVE REDUCED-RANK GP WITH UNCERTAIN INPUTS (RRONIG)

The preceding formulas compute the model with all the datapoints simultaneously. We adapt here the equations to sequential updates. This method is called RRONIG, for Reduced-Rank Online Noisy Input GP.

**Initialization.** $\Gamma_l$ is a vector of size $q$ with initial values all equal to 0 as the initial Gaussian Process is supposed to have a null-vector mean. $\Upsilon_l$ is a matrix of size $q \times q$. Its initial value should verify $\Phi_*\Upsilon_l\Phi_*^\top = \kappa(\mathbf{x}^*, \mathbf{x}^*)$ with $\kappa(\mathbf{x}^*, \mathbf{x}^*)$ the initial covariance chosen for the model (see Solin et al. (2015)).

**Update.** We write above for simplification $\Phi_* = \Phi(\mathbf{x}^*)$, $\Phi = \Phi(\mathbf{x})$ and $\text{vec}(\mathbf{b})$ a vectorization of all the measures ordered by their index. In the following $\Phi_l = \Phi(\mathbf{x}^l)$ with $l$ the index of a datapoint.

For the new datapoints, we compute their associated variance, for the $l$-th datapoint:

$$K_l = \Phi_l\Upsilon_{l-1}\Phi_l^\top + \frac{1}{2}(\Phi_l\Upsilon_{l-1}\zeta_l^\top + \zeta_l\Upsilon_{l-1}^\top\Phi_l) + D_{\text{Var},l} + \Sigma_b. \tag{17}$$

After computing $K_l$, $\Phi_l$ and $\zeta_l$, the parameters $\Upsilon_{l-1}$ and $\Gamma_l$, associated to the expected value and the variance, can be updated:

$$J_l = \Upsilon_{l-1}(\Phi_l + \frac{1}{2}\zeta_l)^\top K_l^{-1} \tag{18}$$

$$\Gamma_l = \Gamma_{l-1} + J_l\Big(vec(\mathbf{b})_l - (\Phi_l + \frac{1}{2}\zeta_l)\Gamma_{l-1}\Big) \tag{19}$$

$$\Upsilon_l = \Upsilon_{l-1} - J_lK_lJ_l^\top. \tag{20}$$

These formulas can be applied iteratively, for every new datapoint, and present the advantage of being applicable to high numbers of data.

**Prediction.** We can write the expected value and covariance in the following form, with $l$ the index of the last datapoint used to update the formulas:

$$\begin{cases} \mathbb{E}[B(\mathbf{p}^*)] \approx \Phi_*\Gamma_l \\ \mathbb{V}[B(\mathbf{p}^*)] \approx \Phi_*\Upsilon_l\Phi_*^\top. \end{cases} \tag{21}$$

Equations 21 predict the value of $B(\mathbf{x}^*)$ and its incertitude on the domain $\Omega$.

**Complexity.** The operations are finding the values of the $q$ eigenfunctions for the studied point ($O(q)$) and the addition and multiplication of vectors of size $q$ with matrices of size $q \times q$ ($O(q^2)$). $K_l$ is a square matrix of the dimension of the output so its inversion is not costly. The operations are repeated $N$ times. The total complexity of the algorithm is then $O(Nq^2)$ for $N$ datapoints and $q$ basis functions, while for classical GPs the complexity scales in $O(N^3)$. In practice, $q$ is fixed and significantly smaller than the number of datapoints, which makes it tractable for high values of $N$. The complexity is the same as for a classical reduced-rank GP (Solin & Särkkä, 2020) and sparse methods (Bijl et al., 2015).

## 5 EXPERIMENTS

### 5.1 COMPARISON ON SIMULATED DATA OF NOISY INPUT GP MODELS

First, we evaluate our model RRONIG on simulated data. The experiments are based on the ones presented in Bijl et al. (2017) in order to compare several GP models that include the input errors. All the experiments (including the ones with real data) were run on a computer with an Intel Core i7-7820HQ CPU. The simulated functions to be interpolated are obtained from the results of a classical Gaussian process. This GP uses the model of the exponential-quadratic covariance: $\kappa_f(x^i, x^j) = \sigma_f^2 exp\left(-\frac{(x^i - x^j)^2}{2l_f^2}\right)$. The values of this Gaussian process are random and follow a normal distribution. The generated function has values in the interval $[-5, 5]$ and is the true function.

For each simulated function, some noisy measures are generated: a noise on the inputs and another one on the outputs is added. 200 artificial measures are generated randomly. These points are the values used for learning by the different models. We study the differences between the predictions from the models and the true function at 100 points, that are homogeneously distributed on the domain of definition. These points are distinct from the ones used for training.

SONIG (Sparse Online Noisy Input Gaussian Process, Bijl et al. (2017)) and RRONIG, our method, are two GP algorithms for online execution. For fair comparison, we used 21 nodes for SONIG and 21 eigenfunctions for RRONIG. Because they iteratively update their model, one measurement at a time, they can not be easily adapted to present the capability to estimate the hyperparameters from all the measures at once. In order to compare the different algorithms in the same conditions, we use for both of our models the hyperparameters computed by NIGP (Noisy Input GP, Mchutchon & Rasmussen (2011)) a method including noisy inputs and that estimates the values of the four hyperparameters, $\sigma_f$, $l_f$, $\sigma_x$ (the standard deviation of the input noise) et $\sigma_y$ (the standard deviation of the output noise). Indeed, if each model had its own tuning of hyperparameters, it would have been difficult to know if one model performed better because of its hyperparameters or not.

Table 1: Mean results of our algorithm RRONIG, SONIG and NIGP (Bijl et al., 2017) for 500 generated functions. For SONIG, the functions for which SONIG diverges are excluded. The criterion to select them is a mean squared error higher than 0.3. There are 2 in the first experiment, 7 in the second and 18 in the third. One invalid SONIG function is also excluded in the second and 18 in the third.

| Input standard deviation | Other hyper-parameters | Method | Mean squared error ($\times 10^{-3}$) | Mean variance ($\times 10^{-3}$) | Mean ratio | NLPD |
|---|---|---|---|---|---|---|
| $\sigma_x = 0.2$ | $\sigma_y = 0.05$ | NIGP | $4.53 \pm 3.02$ | $1.94 \pm 0.93$ | $2.43 \pm 0.86$ | $-1.20 \pm 0.48$ |
| (reestimated by NIGP) | $\sigma_f = 1$ $l_f = 1$ | SONIG (excluding its failures) | $4.03 \pm 6.02$ | $2.41 \pm 1.34$ | $1.54 \pm 0.84$ | $-1.53 \pm 0.47$ |
| | | RRONIG | $\mathbf{3.65 \pm 2.65}$ | $2.32 \pm 1.13$ | $\mathbf{1.45 \pm 0.73}$ | $\mathbf{-1.56 \pm 0.38}$ |
| $\sigma_x = 0.3$ | $\sigma_y = 0.2$ | NIGP | $27.59 \pm 13.37$ | $8.48 \pm 2.99$ | $3.33 \pm 1.16$ | $0.11 \pm 0.59$ |
| (reestimated by NIGP) | $\sigma_f = 1$ $l_f = 0.8$ | SONIG (excluding its failures) | $25.34 \pm 22.89$ | $14.16 \pm 8.11$ | $1.74 \pm 0.91$ | $\mathbf{-0.47 \pm 0.48}$ |
| | | RRONIG | $\mathbf{22.08 \pm 11.48}$ | $13.86 \pm 6.52$ | $\mathbf{1.69 \pm 1.01}$ | $\mathbf{-0.47 \pm 0.44}$ |
| $\sigma_x = 0.4$ | $\sigma_y = 0.1$ | NIGP | $26.72 \pm 15.70$ | $5.46 \pm 2.75$ | $5.19 \pm 1.66$ | $0.76 \pm 0.24$ |
| (reestimated by NIGP) | $\sigma_f = 1$ $l_f = 1$ | SONIG (excluding its failures) | $23.10 \pm 24.12$ | $9.36 \pm 6.85$ | $2.31 \pm 1.37$ | $\mathbf{-0.46 \pm 0.85}$ |
| | | RRONIG | $\mathbf{21.18 \pm 15.19}$ | $10.38 \pm 6.54$ | $\mathbf{2.17 \pm 1.54}$ | $-0.40 \pm 0.70$ |

We now evaluate the methods based on the three essential criteria for practical usage, that we stated in our introduction: accuracy of the prediction (and of its associated variance), robustness and speed. The results are presented in the Table 4. The size of the modeling domain is chosen 1.2 times larger than the real domain of definition in order to limit the effect of the Dirichlet boundary conditions (Solin & Särkkä, 2020; Riutort-Mayol et al., 2022). For the three sequences of the table, the mean squared error on the predictions with SONIG has decreased by 10.9% compared to NIGP when cases of divergence are excluded from the results; with RRONIG it decreased by 20.0%. It indicates that the predictions from RRONIG are significantly better than the ones from SONIG and NIGP.

The RRONIG predictions are more accurate than SONIG but also its predicted variances are considerably closer to the desired values. The variance represents the amplitude of the errors of the model that should be expected: it should be equal to the mean squared error. If the ratio of the error over the variance is higher than 1, the model is too confident in its predictions compared to its actual performances: numerous parts of the real function will be outside of its 95% certainty region. Our method, RRONIG, has a smaller ratio for all the studied cases, not exceeding 2.2, whereas the ratio for SONIG is equal to 2.3 when $\sigma_x = 0.4$ and $\sigma_y = 0.1$ if only some sequences are selected, knowing which predictions are too far from the ground truth. The Figure 2 presents an example of the predictions of SONIG and RRONIG on the same data. The 95% certainty region is larger for RRONIG than for SONIG: the true function is in the certainty region for almost all values of $x$. For SONIG, in the interval $[-5; -4]$ or $[1.3; 2]$, the predictions are outside the certainty region, whereas it is not the case for RRONIG, for which the ground truth is still inside the envelope, even if close to its border.

Concerning the robustness, we observe that SONIG sometimes diverges: the obtained prediction is far from the ground truth. With $\sigma_x = 0.4$ and $\sigma_y = 0.1$, SONIG is considered as invalid for 18 experiments over 500 and 17 of them return results that do not seem to match with the training set (see Figure 1 for an illustration of this divergence). The derivatives calculations by SONIG are affected by noise while they are considered as exact by the model, leading sometimes to the apparition of high gradients in the functions. This is probably the main reason for divergence. We did not encounter this difficulty for any of the test sequences with NIGP and RRONIG. NIGP does not present this issue probably because it uses only one approximation (the Taylor decomposition): its formulas are exact, and all the datapoints are learned at the same time, smoothing the gradients that would have been caused by an isolated datapoint used for learning. For RRONIG, the basis of sine functions seems to mitigate this problem, taking into account the whole profile of the function and not only localized disturbances, thus making it more robust.

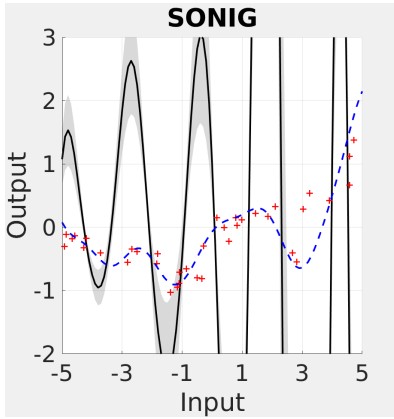

Figure 1: Example of a divergence of the predictions by SONIG (Bijl et al., 2017) with $\sigma_x = 0.4$ et $\sigma_y = 0.1$. The red crosses indicate the measures. The predicted function is the black straight line and the 95% certainty region is in grey. In dashed blue, the true function that we aim to approximate.

As for the execution times (evaluated in Matlab for all algorithms), the average time for learning for SONIG is 337.8 ms (mean for 100 different functions) while for RRONIG it is 6.8 ms, using an Intel Core i7-7820HQ CPU (see Table 2). Our algorithm has a faster computing time, due in particular to the fact that no reestimation of the input noise variance is necessary and only a few matrix computations is required, while numerous different matrices are computed and multiplied for

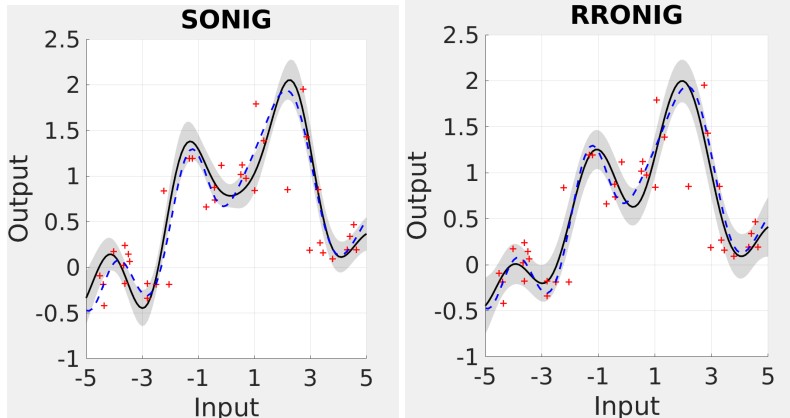

Figure 2: Illustrative example of the predictions by SONIG (Bijl et al., 2017), on the left, and our method (RRONIG), on the right with $\sigma_x = 0.4$ and $\sigma_y = 0.1$. For clarity, we plotted only 40 measures out of the 200 as red crosses. The predicted function is the black straight line and the grey area corresponds to the 95% certainty region. The blue dashed line represents the latent function that we aim to approximate. We observe a better agreement of the truth with RRONIG as well as a better inclusion in the 95% certainty region.

Table 2: Execution time of two online algorithms, SONIG and RRONIG, learning from 200 datapoints and for prediction of the function at 100 different coordinates (mean on 500 different executions).

| Algorithm | Learning time (ms) | Prediction time (ms) |
|-----------|--------------------|----------------------|
| SONIG     | 337.8              | 0.487                |
| RRONIG    | **6.8**            | **0.196**            |

SONIG. The prediction times are much smaller than the learning times with both methods and we can therefore consider prediction cost as negligible compared to the learning cost.

## 5.2 MAGNETIC MAP CREATION AND ONLINE LOCALIZATION WITH REAL DATA

RRONIG has also been tested on real SLAM problems, where input errors (*i.e.* position errors) are a recurrent difficulty for accurate mapping and consequently localization in the mapped areas. The algorithm we used is an online magneto-visual-inertial Multi-State Constraint Kalman Filter (MSCKF). It is an algorithm of the state of the art for online localization that relies on maps of the magnetic field to improve its position estimation (Coulin et al., 2022). In these experiments the magnetic maps are created online, using Gaussian processes, as the device is exploring the area. The inputs for the GP are the positions (corrupted by noise) and the outputs the magnetic map.

The magnetic field is a 3D-vector field. For its modeling, other constraints specific to its properties can be added. As mentioned in Solin et al. (2015), under the assumption that the free-current is negligible, the magnetic field is curl-free and thus is the gradient of a scalar potential, $\varphi$. $\varphi$ can be modeled as a single Gaussian process. As in Equation 4, its variance $\kappa_\varphi$ can be written as: $\kappa_\varphi(\mathbf{x}^\star, \mathbf{x}) = \phi(\mathbf{x}^\star)\Lambda\phi(\mathbf{x})^\top$. As the covariance is a bilinear function, we can obtain the covariance of $B$, the magnetic field, from $\kappa_\varphi$:

$$\kappa_B(\mathbf{x}^\star, \mathbf{x}) = \nabla\phi(\mathbf{x}^\star)\Lambda\nabla\phi(\mathbf{x})^\top. \qquad (22)$$

The expression of the initial $\Lambda$ is $diag(\sigma_{lin}^2, \sigma_{lin}^2, \sigma_{lin}^2, S_{SE}(\lambda_1), ..., S_{SE}(\lambda_q))$ with $d = 3$. The equations are the same as in the Section 4.2, but $\Phi$ should be replaced by $\nabla\Phi$.

We run the magnetic-field based SLAM with magnetic map construction to evaluate the quality of our modeling. As the method should be applicable online and requires to be robust, RRONIG is compared only to online classical GP methods. The real sensor noise (or output noise) is $\sigma_{noise} = 0.3\mu$T. A

common approach to take into account localization errors is to increase artificially the output noise for the modeling (Kok & Solin, 2018; Coulin et al., 2022): we therefore run the GP models with increased noise ($\sigma_m = 4.47\mu$T). It is a coarse instance of the methods described in the first paragraph of the Related work. We used several sequences: two of them consisted in a lot of loops around a table or some chairs (Sequences 1 and 2 in the Table 3). The third Sequence consists in going back and forth along some corridors.

To compare the executions of our algorithms we introduce the Absolute Translation Error (ATE), defined as $e = \sqrt{\frac{1}{N}\sum_{j=1}^{N}||\mathbf{t}_{GI}^j - \hat{\mathbf{t}}_{GI}^j||^2}$ with $N$ the number of data. $\mathbf{t}_{GI}$ is the ground truth position and $\hat{\mathbf{t}}_{GI}$ its associated online estimation.

Our method, which includes the position uncertainty in the GP estimation of the magnetic field increases significantly the robustness of the algorithm. For three sequences, the trajectory diverges after a short period of time with a classical reduced-rank GP when $\sigma_m = \sigma_{noise}$. With the algorithm we proposed, this phenomenon is not observed and there is a high gain in accuracy compared to previous methods. The ATE is improved by 25.5% compared with the naive solution consisting in setting $\sigma_m = 4.47\mu$T. As the localization error is not constant along the trajectory but tends to increase, our model is better adapted for the map creation and is more coherent with the physical phenomenon than modifying sensor noise.

Table 3: ATE of online magneto-visual-inertial SLAM algorithm (1 camera) with different models for magnetic map update. The results of visual-inertial odometry are also indicated.

| Sequence number | Sequence 1 (5 min 44, 288 m) | Sequence 2 (6 min 26, 346 m) | Sequence 3 (4 min 29, 248 m) |
|---|---|---|---|
| Visual-inertial odometry | 1.760 m | 1.806 m | 0.736 m |
| Magneto-visual-inertial MSCKF $\sigma_m = 0.3\mu$T, Coulin et al. (2022) | divergence | divergence | 3.709 m |
| Magneto-visual-inertial MSCKF $\sigma_m = 4.47\mu$T, Coulin et al. (2022) | 1.540 m | 1.179 m | 0.565 m |
| Magneto-visual-inertial MSCKF $\sigma_m = 0.3\mu$T, Our method | **1.017 m** | **0.894 m** | **0.461 m** |

## 6 CONCLUSION

To address the issue of field modeling with input noise, methods that possess solid accuracy, robustness and speed are required. For this aim, we proposed an online reduced-rank method with Taylor expansion named RRONIG. Fewer steps and calculations are required compared to the previously existing online method, SONIG. On simulated data that is standard for comparison, this algorithm shows a significant improvement of the precision and computational time over the existent methods. Moreover, it does not suffer from divergence unlike the other online method, making it also a more robust candidate. Following these observations, we investigated the performance of our method on a challenging practical task, an application of SLAM, which also suffers from noisy inputs. Our results show an improvement over the state of the art on this context.

For future work, it would be interesting to investigate further the reasons of the robustness of our algorithm compared to the others, as it could be a useful guidance for future design of algorithms. Another interesting extension of our work would be to evaluate the approximation errors of the derivatives due to the lack of knowledge of the ground truth, as we expect it to further ameliorate the estimation of the confidence intervals of the predictions of our algorithm. Finally, most sensors used for data collection are noisy, thus an accurate modeling of the noise on both the inputs and the outputs may be essential: this algorithm could be applied to various domains as we did in this paper for localization.

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

# A  APPENDIX

$K_2$ and $K_3$ depend on the distribution of $B_{IE}$ and $B$. We determine the elements of $K_2$ first with

$$\text{Cov}\Big(B(\mathbf{x}^\star), B(\mathbf{x}+\xi)\Big) \simeq \text{Cov}\Big(B(\mathbf{x}^\star), B(\mathbf{x}) + \nabla B(\mathbf{x})\xi + \frac{1}{2}h(B(\mathbf{x}),\xi)\Big). \tag{23}$$

The covariance is linear, therefore

$$\text{Cov}\Big(B(\mathbf{x}^\star), B(\mathbf{x}+\xi)\Big) \simeq \text{Cov}\Big(B(\mathbf{x}^\star), B(\mathbf{x})\Big) + \text{Cov}\Big(B(\mathbf{x}^\star), \nabla B(\mathbf{x})\xi\Big) + \text{Cov}\Big(B(\mathbf{x}^\star), \frac{1}{2}h(B(\mathbf{x}),\xi)\Big). \tag{24}$$

By independence of $B(\mathbf{x})$ and $\xi$ and as $\mathbb{E}[\xi] = \mathbf{0}_d$,

$$\text{Cov}\Big(B(\mathbf{x}^\star), \nabla B(\mathbf{x})\xi\Big) = \mathbb{E}[B(\mathbf{x}^\star)\xi] - \mathbb{E}[B(\mathbf{x}^\star)]\mathbb{E}[\xi] = 0. \tag{25}$$

For all $j,k$, with $j \neq k$, $\xi_j\xi_k$ and $\frac{\partial^2 B(\mathbf{x})}{\partial \mathbf{x}_j \partial \mathbf{x}_k}$ are independent so

$$\text{Cov}\Big(B(\mathbf{x}^\star), h(B(\mathbf{x}),\xi)\Big) = \sum_{j=1}^d \sum_{k=1}^d \mathbb{E}[\xi_k\xi_j]\Big(\mathbb{E}\Big[B(\mathbf{x}^\star)(\frac{\partial^2 B(\mathbf{x})}{\partial \mathbf{x}_j \partial \mathbf{x}_k})^\top\Big] - \mathbb{E}\Big[B(\mathbf{x}^\star)\mathbb{E}\Big[\frac{\partial^2 B(\mathbf{x})}{\partial \mathbf{x}_j \partial \mathbf{x}_k}\Big]^\top\Big]\Big).$$

As $\mathbb{E}[\xi] = \mathbf{0}_d$, $\mathbb{E}[\xi_j\xi_k] = \text{Cov}(\xi_j, \xi_k)$ and $\text{Cov}(\xi_j, \xi_k) = \text{Cov}(\xi_k, \xi_j)$ by definition, we can write

$$\text{Cov}\Big(B(\mathbf{x}^\star), h(B(\mathbf{x}),\xi)\Big) = \sum_{j=1}^d \sum_{k=1}^d \Sigma_{j,k} \frac{\partial^2 \kappa(\mathbf{x}^\star, \mathbf{x})}{\partial \mathbf{x}_j \partial \mathbf{x}_k} = \Big\langle \Sigma, H_2(\kappa)(\mathbf{x}^\star, \mathbf{x})\Big\rangle. \tag{26}$$

The operator $\langle \cdot, \cdot \rangle$ corresponds to the trace of the product of the two matrices generalised to tensors and $H_2$ to the second derivative along the second component of the covariance ($H_1$ is for the second derivative along the first element).

For the covariance $K_3$, we study the covariance for two mesures at $\mathbf{x}^\alpha$ and $\mathbf{x}^\nu$, with $\alpha$ and $\nu$ the measures' respective indices and $\kappa(\mathbf{x}^\alpha, \mathbf{x}^\nu) = \text{Cov}(B(\mathbf{x}^\alpha), B(\mathbf{x}^\nu))$,

$$\text{Cov}\Big(B(\mathbf{x}^\alpha + \xi^\alpha), B(\mathbf{x}^\nu + \xi^\nu)\Big)$$
$$\simeq \text{Cov}\Big(B(\mathbf{x}^\alpha) + \nabla B(\mathbf{x}^\alpha)\xi^\alpha + \frac{1}{2}h(B(\mathbf{x}^\alpha),\xi^\alpha)), B(\mathbf{x}^\nu) + \nabla B(\mathbf{x}^\nu)\xi^\nu + \frac{1}{2}h(B(\mathbf{x}^\nu),\xi^\nu)\Big). \tag{27}$$

Separating each term, we obtain

$$\text{Cov}\Big(B_{IE}(\mathbf{x}^\alpha), B_{IE}(\mathbf{x}^\nu)\Big) \simeq \kappa(\mathbf{x}^\alpha, \mathbf{x}^\nu) + \delta_{\alpha,\nu} \times (\Sigma_b + \text{Var}(\nabla B(\mathbf{x}^\alpha)\xi^\alpha))$$
$$+ \frac{1}{2}\Big\langle \Sigma^{t^\alpha}, H_1(\kappa)(\mathbf{x}^\alpha, \mathbf{x}^\nu)\Big\rangle + \frac{1}{2}\Big\langle \Sigma^{t^\nu}, H_2(\kappa)(\mathbf{x}^\alpha, \mathbf{x}^\nu)\Big\rangle. \tag{28}$$

$\delta_{\alpha,\nu}$ is the Kronecker symbol: if and only if $\alpha = \nu$, the sensor's and the position white noises are not independent, according to our hypothesis, since they are the same ($\xi_\alpha = \xi_\nu$ and $\varepsilon_\alpha = \varepsilon_\nu$).

The variance obtained for $K_3$ is different from the one proposed in Bijl et al. (2017): the term $\frac{1}{2}\Big\langle \Sigma^{t^\alpha}, H_1(\kappa)(\mathbf{x}^\alpha, \mathbf{x}^\nu)\Big\rangle$ is present but also its symmetric counterpart, due to the fact that both variables, $B_{IE}(\mathbf{x}^\alpha)$ and $B_{IE}(\mathbf{x}^\nu)$, are corrupted by input noise. This supplementary term ensures the symmetry of the variance matrix when $\mathbf{x}^\alpha = \mathbf{x}^\nu$. Knowing the expressions of $K_2$ and $K_3$, the System 7 can be solved numerically. With the reduced rank approximation the different components of $K_2$ and $K_3$ can be written differently with

$$\Big\langle \Sigma, H_\mathbf{x}(\kappa)(\mathbf{x}^\star, \mathbf{x})\Big\rangle = \Big\langle \Sigma, H_\mathbf{x}(\phi_\star \Lambda \phi^\top)\Big\rangle = \sum_{j=1}^d \sum_{k=1}^d \Sigma_{j,k} \frac{\partial^2 \phi_\star \Lambda \phi^\top}{\partial \mathbf{x}_j \partial \mathbf{x}_k}. \tag{29}$$

$\phi_\star \Lambda$ is independent of the variable $\mathbf{x}$ so

$$\Big\langle \Sigma, H_\mathbf{x}(\kappa)(\mathbf{x}^\star, \mathbf{x})\Big\rangle = \sum_{j=1}^d \sum_{k=1}^d \Sigma_{j,k} \phi_\star \Lambda \frac{\partial^2 \phi^\top}{\partial \mathbf{x}_j \partial \mathbf{x}_k}. \tag{30}$$

As $\Sigma_{j,k}$ is a scalar value,

$$\left\langle \Sigma, H_{\mathbf{x}}(\kappa)(\mathbf{x}^\star, \mathbf{x}) \right\rangle = \phi_\star \Lambda \sum_{j=1}^{d} \sum_{k=1}^{d} \Sigma_{j,k} \frac{\partial^2 \phi^\top}{\partial \mathbf{x}_j \partial \mathbf{x}_k} = \phi_\star \Lambda \zeta \quad \text{with} \quad \zeta = \sum_{j=1}^{d} \sum_{k=1}^{d} \Sigma_{j,k} \frac{\partial^2 \phi^\top}{\partial \mathbf{x}_j \partial \mathbf{x}_k}. \tag{31}$$

$\zeta$ is of length $q$ (for a unique point $\mathbf{x}$). We can write the explicit expressions of the matrices studied,

$$K_2 \simeq \begin{bmatrix} \phi_\star \Lambda \phi^\top + \frac{1}{2} \phi_\star \Lambda \zeta \\ \phi_\diamond \Lambda \phi^\top + \frac{1}{2} \phi_\diamond \Lambda \zeta \end{bmatrix} \quad \text{and} \quad K_3 \simeq \left[ \phi \Lambda \phi^\top + \frac{1}{2} \left( \phi \Lambda \zeta^\top + \zeta \Lambda \phi^\top \right) + D_{\text{Var}} + \Sigma_b \right] \tag{32}$$

with $D_{\text{Var}} = \text{Var}\left( \nabla B(\mathbf{x}) \xi \right)$.

Finally, for several values, the equations can be generalized as:

$$\begin{cases} \mathbb{E} \left[ B(\mathbf{x}^{\star 1}, ..., \mathbf{x}^{\star p}) | \mathbf{b}_1, ... \mathbf{b}_N \right] = \mu_{B_\star} + K_2 K_3^{-1} vec(\mathbf{b}^1, ... \mathbf{b}^N) \\ \text{Cov}\left( \begin{bmatrix} B(\mathbf{x}^\star) \\ B(\mathbf{x}^\diamond) \end{bmatrix}, \begin{bmatrix} B(\mathbf{x}^\star) \\ B(\mathbf{x}^\diamond) \end{bmatrix} | \mathbf{b} \right) = \text{Cov}\left( \begin{bmatrix} B(\mathbf{x}^\star) \\ B(\mathbf{x}^\diamond) \end{bmatrix}, \begin{bmatrix} B(\mathbf{x}^\star) \\ B(\mathbf{x}^\diamond) \end{bmatrix} \right) - K_2 K_3^{-1} K_2^\top \end{cases} \tag{33}$$

with

$$\begin{cases} K_2 = \left( \Phi_\star \Lambda \Phi^\top + \frac{1}{2} \Phi_\star \Lambda \Theta^\top \right) \\ K_3 = \left( \Phi \Lambda \Phi^\top + \frac{1}{2} \left( \Phi \Lambda \Theta^\top + \Theta \Lambda \Phi^\top \right) + D_{\text{Var}} + \Sigma_b \right). \end{cases} \tag{34}$$

and

$$D_{\text{Var}} = \begin{bmatrix} \text{Var}\left( \nabla B(\mathbf{x}_1) \xi_1 \right) & 0 & \mathbf{0}_{1 \times (n-2)} \\ 0 & \text{Var}\left( \nabla B(\mathbf{x}_2) \xi_2 \right) & \mathbf{0}_{1 \times (n-2)} \\ ... & ... & ... \end{bmatrix}. \tag{35}$$

FURTHER RESULTS

In order to test the algorithm on simulated datasets with heteroscedastic noise variance, we created a simulated dataset similar to the previous ones, but this time the input noise is regularly increasing: we decided to define its standard deviation as $0.4 \times n/N$, with n the index of each new datapoint incorporated and N the total number of datapoints. For SONIG and RRONIG, for each new update, we gave to the model the value of its corresponding input noise standard deviation. Unfortunately, it is more difficult to modify NIGP to include heteroscedastic noise: it uses a constant value of $\sigma_x$, tuned by its hyperparameter optimization step. Our method performs better than the others (see Table 4).

Table 4: Mean results of our algorithm RRONIG, SONIG and NIGP (Bijl et al., 2017) for 500 generated functions. For SONIG, the functions for which SONIG diverges are excluded. There is one invalid SONIG function in this experiment.

| Input standard deviation | Other hyper-parameters | Method | Mean squared error ($\times 10^{-3}$) | Mean variance ($\times 10^{-3}$) | Mean ratio | NLPD |
|---|---|---|---|---|---|---|
| $\sigma_x = 0.4n/N$ | $\sigma_y = 0.1$ | NIGP | $6.76 \pm 3.99$ | $3.03 \pm 1.29$ | $2.25 \pm 0.91$ | $-0.99 \pm 0.34$ |
| (reestimated by NIGP) | $\sigma_f = 1$ $l_f = 1$ | SONIG (excluding its failures) | $3.45 \pm 2.30$ | $2.67 \pm 1.27$ | $1.29 \pm 0.63$ | $-1.52 \pm 0.49$ |
| | | RRONIG | $\mathbf{3.07 \pm 2.03}$ | $2.46 \pm 1.03$ | $\mathbf{1.22 \pm 0.78}$ | $\mathbf{-1.56 \pm 0.34}$ |

