# OpenReview forum: "Reduced-Rank Online Gaussian Process Modeling With Uncertain Inputs"
_ICLR.cc/2024/Conference — Submitted to ICLR 2024_

### Official Review · Reviewer_37Cr · 2023-10-23

**Soundness:** 2 fair
**Presentation:** 1 poor
**Contribution:** 2 fair
**Rating:** 3
**Confidence:** 4

**Summary:**

This paper introduces an approach to online (multi-output) Gaussian process inference for uncertain inputs using a reduced-rank approximation of the covariance matrix for efficiency. The approach is based on approximating the latent function evaluated on a corrupted input via a second-order Taylor approximation. The proposed method is evaluated experimentally on a synthetic problem and on an indoor localization task based on measurements of the ambient magnetic field.

**Strengths:**

The paper proposes a simple idea to handle uncertain inputs by Taylor expanding the GP evaluated on noisy inputs and combines this formulation with a low rank formulation based on a truncated series of the covariance function. This formulation promises computational speedups and initial results on a synthetic dataset that the authors show seem promising.

**Weaknesses:**

While the paper combines existing ideas to address the challenges of online GP regression with uncertain inputs, there are significant weaknesses in how the results are presented and evaluated:

## Originality
The paper relies on existing ideas for low-rank Gaussian processes (Solin & Särkkä, 2020) for scalability and builds on approaches on how to incorporate noisy data via a Taylor approximation. At times it is difficult to separate what this paper adds from what is known. In particular, how the Taylor approximation differs from the one considered by Bijl et al, 2017. The fact that GPs can be updated in an online fashion through (tractable) sequential conditioning is also well known (e.g. via updating a Cholesky factorization).

## Method and Theoretical Results
The method is explained in a lengthy fashion but at crucial points leaves out details. For example, as above, what is new about the use of the Taylor approximation? Also how do the update equations (20) - (23) arise? Are these just based on the standard sequential view of GP inference? If yes, why is that a novel contribution? If no, how are they motivated / derived?
The online formulation of the noisy-input GP is the main contribution, so this seems like a crucial point.

## Experimental Evaluation
The experimental evaluation is unfortunately very limited. In only compares to other online approaches on a synthetic dataset using an RBF kernel. There is no consideration of how to choose the rank of the approximation or how to optimize hyperparameters.  It was also unclear to me whether the baselines were also implemented in C++ as the author's approach or in Matlab, which potentially can lead to significant runtime differences.

### Simulated Data
For the simulated data a more thorough comparison on different kernels in my opinion is necessary, in particular because low-rank approximation will work especially well on the chosen exponential kernel due to its fast decaying spectrum. While results in Table 1 seem to be promising, the differences in performance are within one standard deviation, making it hard to judge whether they are systematic or an artefact. Also there is no systematic variation of hyperparameters: It is unclear how the choices of "other hyperparameters" were made.

### Magnetic Map Creation
The only other experiment, the magnetic map creation problem, does not compare RRONIG to another noisy-input method as far as I can tell, which I do not understand. This does not seem an informative comparison to me. If the reduced-rank approach for noisy-inputs is the main contribution, it would seem a comparison to SONIG is required here, no matter its robustness. In fact, if SONIG turns out to be non-robust on real data that would seem to be a benefit of RRONIG to be showcased. Further, error bars in Table 3 are missing, making it impossible to judge how large the performance difference is across different runs.

### Suggested experiments
To improve the paper I would recommend some systematic ablation experiments on (synthetic) data. For example, choosing Matern kernels of increasing smoothness to judge the effectiveness of a low-rank approximation and the required rank for good performance. An experiment to evaluate the impact of the choice of rank or a strategy to choose the rank. Finally, an ablation that considers model mismatch, i.e. how RRONIG performs under slightly misspecified hyperparameters, which will be relevant in practice, since as far as i can tell no model selection strategy is presented.

## Quality of Presentation / Clarity
The quality of presentation and clarity of the paper could be much improved. There is a lot of atypical language usage which makes the paper hard to understand in parts and can lead to confusion with existing mathematical terminology.

Examples:
- "modelize" => model
- "measure" (measurement?) versus (probability) measure
- the "measure is ignored" => measurement is ignored?
- "truth function" => true / latent function
- "conditionned to" => conditioned

Further, the paper could be simplified to make space for better visual illustrations of the results and the above ablation experiments. For example eqn 4 and 23 give the identical formula of the exponentiated quadratic kernel, which is well-known.

## Minor Comments Not Affecting Score
- Paper uses the ICLR 2023 template, not the 2024 template.
- Code comments still state this paper is being submitted to NeurIPS 2023.

**Questions:**

- How do the update equations (20) - (22) arise? There is no explanation or derivation for them as far as I can tell.
- How do you choose the number of eigenfunctions for the low-rank approximation? Is there an automatic way or is it fixed by hand?
- How do you estimate kernel hyperparameters with your method? In the synthetic experiments you seem to use NIGP to do so.
- What do you mean by "more complete version of the Taylor approximation" compared to Bijl et al. (2017)? What are the exact differences to the approach by Bijl et al (2017)?
- Why are you only comparing your (online) noisy input approach to approaches which are not designed for noisy inputs (such as SONIG) on the magnetic map creation dataset?

---

> ### Author Response · Authors · 2023-11-20
> **Part 1**
>
> We thank the reviewer for their time and their detailed and constructive feedback! Please find detailed responses in the following.
>
> $\textbf{Originality}$
>
> Concerning the contribution of our paper,  we rewrote a part of the Section 1 as follows to make it clearer. Please see the beginning of the response to reviewer 1 for more details.
>
> $\textbf{Experimental evaluation}$
>
> As you pointed out, it is indeed known that low rank approximation is more efficient for Gaussian kernel than for example Matern kernels, as the latter requires larger rank to efficiently approximate the inference of a given kernel [Daskalakis2022]. Our method is done for a RBF kernel only. However, we don’t consider it necessarily as a drawback as it is the most commonly used kernel. Moreover, we consider the study of other kernels to be out of the scope of our paper, which addresses the problem of absence of a method that combines efficiency, speed and robustness.
>
> Please see our response number 2 to the second reviewer for the choice of kernel hyperparameters.
>
> Our method is implemented in both C++ and Matlab. We will further emphasize in Section 5.1 that the comparison in terms of performance and execution time were made in Matlab using an Intel Core i7-7820HQ CPU for all the algorithms for a fair comparison. Our method is also coded in C++, which was used in Section 5.2, as it is more efficient and, thus, is the one we put in the supplementary material and we plan to make public. If the reviewer considers of interest that we also put the Matlab version, we will be very happy to.
>
>
> $\textbf{Suggested experiments}$
>
> We think that choosing different kernels could be an interesting perspective to broaden our understanding of the behaviour of reduced-rank GPs but we consider that it would be more appropriate in another paper as our focus was on RBF kernels.  Under slightly misspecified hyperparameters, RRONIG (and the other models as well) do not show difficulties in convergence, apart from the ones already mentioned for SONIG. The hyperparameters used for learning are the ones returned by the hyperparameter optimization of NIGP and not the ground truth hyperparameters so we can consider them as slightly mismatched. Moreover, one of the secondary purposes of our real case experiment is to show the capabilities of our method with approximate hyperparameters as they are not updated for each experiment, while the settings between the three of them are different (not the same places are explored and the paths are different).
>
> We took into account your grammatical remarks in the paper and thank you for them.

---

> ### Author Response · Authors · 2023-11-20
> **Part 2**
>
> $\textbf{Questions}$
>
> 1) The equations (18)-(22) are based on the updates of the data in online GP.  They follow the steps used in the book "Bayesian Filtering and Smoothing" [Sarkka2013] to transform batch updates to a sequential algorithm, adapted to our own equations.
>
> 2) The number of eigenfunctions chosen depends on the lengthscale of the function variations and the width of the studied interval. We tried to find the best compromise between a low number of functions and enough accuracy in the predictions. For further details, [Daskalakis2022] gives theoretical bounds for the rank estimation for an efficient approximation of Gaussian kernels.
>
> 3) Yes, we used hyperparameters obtained from NIGP and we explain why in the response 2 to the second reviewer.
>
> 4) Our method is different from the one used by Bijl et al. in three different ways:
> - we do not re-estimate the distribution of the input noise, reducing the number of steps needed for the algorithm execution
> - using the Taylor approximation, our development is slightly different as one of the second order terms (the third term of Equation (16)) seems to have been omitted in (17) in Bijl et al.
> - we formulated all the update equations for prediction using a reduced-rank approximation instead of a sparse method, in order to address the lack of robustness of SONIG (even when correcting the omission discussed in the previous point). The reduced-rank approximation is motivated by the fact that it enables to use a kernel of smooth eigenfunctions, with the aim of mitigating the divergence issues.
>
> 5) The other experiment, the Magnetic Map Creation, is a challenging problem and was done with real data. It does not consist only in learning a GP model, the GP algorithm here is in constant interaction with other parts of the SLAM algorithm. Adapting existing GP methods is non-trivial and no method with input noise has been proposed for this task to the best of our knowledge. The purpose of the experiment is two-folds: the relevance of taking into account the input noises and the simplicity and adaptability of our method that allows a successful implementation for this problem.
>
> SLAM acquisitions are complex (with a necessary initialization phase that is different for each run), as well as costly in time. Given the very complicated task to reproduce the same run, we consider that averaging over the different runs does not make sense.
>
> In Table 1, the large standard deviations are due to the fact that there can be a high variability of the quality of the GP predictions depending on the functions and the learning points (the input and the output noise are random and we learn with 200 points only, which create a high variability between the runs). As suggested by another reviewer, we added the NLPD values in order to prove that our algorithm performs better in general.
>
> [Daskalakis2022] Daskalakis, Constantinos, Petros Dellaportas, and Aristeidis Panos. "How good are low-rank approximations in Gaussian process regression?." Proceedings of the AAAI Conference on Artificial Intelligence. Vol. 36. No. 6. 2022.
>
> [Bijl2017] H. Bijl, T. B. Schön, J.-W. van Wingerden, and M. Verhaegen. System identification through online sparse gaussian process regression with input noise. IFAC Journal of Systems and Control, 2:1–11, 2017
>
> [Sarkka2013] Simo Sarkka (2013). Bayesian Filtering and Smoothing. Cambridge University Press.

---

> > ### Comment · Reviewer_37Cr · 2023-11-22
> >
> > Thank you for your response. My assessment remains unchanged.

---

### Official Review · Reviewer_9wSx · 2023-10-25

**Soundness:** 4 excellent
**Presentation:** 3 good
**Contribution:** 3 good
**Rating:** 6
**Confidence:** 4

**Summary:**

The paper proposes a novel online Gaussian Process modeling for vector field mapping with inputs that are corrupted by noise.
The approach is based on a second order Taylor approximation of the data modeling. The author provide the formulas of the posterior distribution of the GP in the "static" setting (with all the datapoints simultaneously) and online setting (one data point at a time).
Numerical experiments show that the model is robust, fast and accurate compared to the most recent online GP with uncertain inputs.

**Strengths:**

Overall I have to say that it was pleasant to read the paper.
Strengths of the paper:
    - The paper is well written. It is easy to read as it is clear from the beginning to the end. The motivations are also very clear and sound.
    - While first order Taylor approximation has already been used in such a context, the authors propose to push this to second order combined with truncated approximation of covariance with smooth eigenfunctions. The derivation is well done when all the data are considered simultaneously.
    - the numerical experiments are relevant and show the benefit of the author's approach

**Weaknesses:**

The paper has a couple of weaknesses.
   - In my opinion, the literature review is not complete. There are many approaches that deal with online regression and sparse approximations (see [1,2]).
   - The paper seems to combine different known facts about uncertain input GPs. Second-order Taylor approximation and smooth basis functions are two tools that have been used already in this context. In other words, I have the feeling that the paper considers the approach of [3] with second-order approximation and usual basis function approximations. Am I right? I am not saying that the contribution is low because of this. However, I think this should be somehow emphasized.
   -  One of the main weakness of the paper is the explanation around equations (18)-(22). This is not straightforward to go from equations in Section 4.1 to Section 4.2. This would have deserved more explanations. Could the authors provide some more in-depth details about equations  (18)-(22) and their links with equations from Section 4.1?
   - If I have not missed it, the paper makes no mention of hyperparameter optimization. This is a challenging a question in an online context. How would the authors incorporate this crucial step in their current approach?
   - It would have been interesting to see some results on different simulated benchmarks datasets (heteroscedastic noise variance for instance)


[1] TD Bui, C Nguyen, RE Turner, Streaming Sparse Gaussian Process Approximations. Advances in Neural Information Processing Systems, 2017
[2] A Gijsberts and G Metta, Real-time model learning using incremental sparse spectrum gaussian process regression. Neural networks, 2013.
[3] A Mchutchon and C Rasmussen Gaussian Process Training with Input Noise. Advances in Neural Information Processing Systems 2011.

**Questions:**

See the weaknesses above.
- Is the code publicly available?

Depending on author responses, I would increase my score.

---

> ### Author Response · Authors · 2023-11-20
>
> We thank the reviewer for the detailed notes and the interesting feedback!
>
> 1. While our related work focused on the incorporation of the input noises, the references you mentioned are indeed relevant and enrich our related work section, we added them to the literature of our paper.
>
> 2. We agree with the fact that the second-order Taylor approximation has already been used in previous GP methods, although with minor omissions of some terms in the development. There are several reduced-rank GP methods in the literature but they do not address the problem of uncertain inputs. We needed to work on creating our own algorithm, basing ourselves on the already existing tools in the literature that we considered of interest. We reformulated our contribution as follows in order to make it clearer:
> In numerous applications, real-time estimation of a model are needed (for example patients’ vital signs or telemetry data, as well as localization). Therefore a GP model that can be used online, with updates of a short computing time for each new measurement, and with a strong reliability (i.e. with a low risk of divergence) is required. The previously existing methods including uncertain inputs did not match these requirements and we created a new GP algorithm for uncertain inputs that includes ideas from a special category of GP modeling, never studied in this context: a method that uses smooth eigenfunctions in order to avoid divergences that could rise. We confirm the relevance of this approach by a comparison on simulated data between our algorithm and other noisy inputs GPs. It showed a better robustness and speed compared to the other existent online algorithm while also having a better accuracy. Moreover, to assert the practical use of this algorithm, we also adapted and evaluated it on a challenging real-life problem, a very active field in robotics: online localization in indoor environments, and obtained better results than the state of the art [Coulin2022].
>
>
> 3. The equations (18)-(22) are based on the updates of the data in online GP.  They follow the steps used in the book
> [Sarkka2013] to transform batch updates to a sequential algorithm, adapted to our own equations.
>
> 4. Please see response 2 to the second reviewer for the hyperparameters optimization.
>
> 5. As you mentioned, it would be interesting to try the algorithm on simulated datasets with heteroscedastic noise variance. We added the results in the supplementary material. We created a simulated dataset similar to the previous ones, but this time the input noise is regularly increasing: we decided to define its standard deviation as 0.4*n/N, with n the index of each new datapoint incorporated and N the total number of datapoints. For SONIG and RRONIG, for each new update, we gave to the model the value of its corresponding input noise standard deviation. Unfortunately, it was more difficult to modify NIGP to include heteroscedastic noise, it uses a constant value of $\sigma\_x$, tuned by its hyperparameter optimization step. Our method performs better than the others (see Table 4). Please note also that on real data, the input noise variance was varying as the localization error estimated from the SLAM algorithm varies over time. RRONIG showed robustness and a strong ability to improve the position's estimation on this dataset as well.
>
> $\textbf{Questions}$
>
> Yes, the code for our algorithm is available as supplementary material and will be public.
>
>
>
> [Coulin2022] J. Coulin, R. Guillemard, V. Gay-Bellile, C. Joly, and A. de La Fortelle. Tightly-coupled magneto-visual-inertial fusion for long term localization in indoor environment. IEEE Robotics and Automation Letters, 7(2):952–959, April 2022
>
> [Sarkka2013] Simo Sarkka (2013). Bayesian Filtering and Smoothing. Cambridge University Press.

---

> > ### Comment · Reviewer_9wSx · 2023-11-22
> > **Thanks for the responses**
> >
> > I thank the authors for the responses and the effort made in addressing the points I raised. If I understand well the contributions, then Equations (17) - (21) are the most important of the paper. In my opinion, this definitinetly deserves explanations because the derivation does not look straightforward for someone who is not familiar with online GP updates.
> > Moreover, like reviewer 1vN1 I think there is lack of numerical experiments to show the numerical stability robustness.
> > It's mainly for these reasons that I keep my score.

---

### Official Review · Reviewer_1vN1 · 2023-10-27

**Soundness:** 2 fair
**Presentation:** 2 fair
**Contribution:** 3 good
**Rating:** 5
**Confidence:** 3

**Summary:**

The authors propose RRONIG, a new online learning approach for GPs in the presence of uncertain inputs. The method is based on a second-order Taylor approximation which results in a reduced rank formulation for the model. The main claims are better accuracy, robustness and shorter computing times when compared to other methods.

**Strengths:**

The work tackles an important problem with practical relevance. Although it has been addressed before for GPs, the better performance with much faster computing time is impressive.

Despite being a bit difficult to follow due to the heavy math notation, the derivations in the main text and in the provided appendix are comprehensive.

**Weaknesses:**

The main weakness of the work is in the experimental section. For the artificial dataset, the proposed RRONIG is compared only against one online method (SONIG, from 2017) and one batch method (NIGP, from 2011). For the real data, only a single model is used for comparison (besides visual-inertial odometry). I believe the experimental results would be more complete with the inclusion of more recent GP methods for uncertain inputs or other online GP methods.

In Section 5.1, the authors did not optimize the kernel hyperparameters in the the RRONIG runs. Is that also true for the experiments in Section 5.2? If so, it seems to be a significant drawback in the overall evaluation, since it is an important aspect of the GP training.

**Questions:**

Below I list a few more comments and questions:

- One should always specify when using the term "robust". Is the proposal robust to model misspecification? To outliers? To numerical instabilities? From the experimental section, it seemed that the robustness is related to better numerical stability, but this should be more clear from the beginning of the paper.

- In Section 2, the title "Modifying GP equations" is too generic. Almost all contributions in the GP literature require changing standard GP equations. Maybe "Input noise modeling" is enough?

- Table 1 should include the log predictive density to better evaluate the models' predicted uncertainties.

- Instead of reporting the average of the predicted variances and the "mean ratio", it would be more direct to simply report the mean log predictive density, which already balances the quality of the predicted means and variances.

- Page 7: "The variance represents the amplitude of the errors of the model that should be expected: it should be equal to the mean error." - Since the variance is in squared units, is the reported error also squared?

- The "reduced-rank GP" competitor considered in the SLAM experiment is the one from Solin et al. (2015)? In the Conclusion, it is claimed "an improvement over the state of the art on this context". Was the proposal by Solin et al. (2015) the state of the art so far for SLAM problems? I believe more recent strategies (even without GPs) should be considered.

---

> ### Author Response · Authors · 2023-11-20
>
> We thank you for your helpful feedback and suggestions!
>
> $\textbf{1, Remarks on the experimental section.}$
> Concerning the methods used for the comparison in the artificial dataset, we did not compare with more recent GP methods with uncertain inputs as there are unfortunately no other more recent methods that are able to incorporate them. To the best of our knowledge SONIG is the state of the art in this context.
>
> Adapting existent GP methods to the use-case of SLAM is non-trivial and must respect certain constraints such as low computational complexity. We wanted to insert the SONIG implementation in this SLAM algorithm for comparison. However, as there are many steps of estimation and numerous matrices to be computed, contrary to RRONIG, we finally decided that the task would be too complicated. We consider the successful implementation of our method with better performance that the state of the art as strong evidence for the simplicity, the adaptability and the capabilities of our algorithm.
>
> $\textbf{2, Remarks on the kernel hyperparameters.}$
> In Table 1, we chose to set the same kernel hyperparameters for the three methods for the purpose of a fair comparison, as done in [1], using the hyperparameters tuned by NIGP. If each model had its own tuning of hyperparameters, it would have been difficult to know if one model performed better only because of better hyperparameters or not. In this paper, it is the quality of the updating step of the different algorithms that we wanted to compare. The hyperparameters' optimization is a complex problem that we consider as outside of the scope of this paper: we place ourselves in the context of online GP learning, studying algorithms able to learn a model even with a few datapoints available at the beginning and improving iteratively. However the hyperparameters can only be deduced when there is a sufficiently high number of datapoints available. The optimization phase of the hyperparameters is a preliminary phase before online learning. This step is also not always required as the choice of hyperparameters in general can be based on expert knowledge (as performed in our SLAM real life application). One alternative if we do not have an idea of the hyperparameters' value would be to perform a fine tuning of the hyperparameters based on a subset of the entries before launching the online algorithm. We updated our paper with more precisions about this.
>
> Note that in our real-life experiment in Section 5.2, we chose the kernel hyperparameters based on expert knowledge. The chosen hyperparameters proved to be very robust when varying the settings. They were the same in all the experiments presented that were conducted in different environments.
>
>
> $\textbf{Questions:}$
>
> 1) Indeed, the lack of robustness originates from numerical instabilities, we added the precision in the beginning of the paper.
>
> 2) We agree with the fact that the suggested title is more explicit, we updated the paper to modify it.
>
> 3) We initially only put the predicted variance and the ratio as we wanted to follow the same evaluation metrics used in [Bijl2017]. However, we agree that the log predictive density metric is relevant and added the results in Table 1.
>
> 4) Thank you for the remark, we added the precision that it is the MSE, mean squared error, that is calculated.
>
> 5) [Coulin2022] is the state of the art for magnetic-based localization. We improved over it and in Section 5.2, our algorithm is compared to executions of [Coulin2022]. In Table 3, we added the reference of the algorithm used on the lines that are concerned. [Solin2015] uses similar equations for magnetic mapping but its SLAM is only magneto-inertial and is not as precise as the one presented in [Coulin2022].
>
> [Bijl2017] H. Bijl, T. B. Schön, J.-W. van Wingerden, and M. Verhaegen. System identification through online sparse gaussian process regression with input noise. IFAC Journal of Systems and Control, 2:1–11, 2017
>
> [Coulin2022] J. Coulin, R. Guillemard, V. Gay-Bellile, C. Joly, and A. de La Fortelle. Tightly-coupled magneto-visual-inertial fusion for long term localization in indoor environment. IEEE Robotics and Automation Letters, 7(2):952–959, April 2022
>
> [Solin2015] A. Solin, M. Kok, N. Wahlström, T. Schön, and S. Särkkä. Modeling and Interpolation of the Ambient Magnetic Field by Gaussian Processes. IEEE Transactions on Robotics, 34(4):1112– 1127, September 2015.

---

> > ### Comment · Reviewer_1vN1 · 2023-11-21
> > **Thanks for the responses**
> >
> > I thank the authors for the detailed responses.
> >
> > I still believe that more online GP (with or without input uncertainty) and non-GP models should be included in the experiments. For instance, it would be possible to consider GPLVM-based models or sampling-based methods for uncertainty propagation. I also think that the kernel hyperparameter learning step deserves more attention in the experimental section.
> >
> > Thus, for now I choose to maintain my rating.

---

### Official Review · Reviewer_RsmZ · 2023-10-31

**Soundness:** 2 fair
**Presentation:** 2 fair
**Contribution:** 2 fair
**Rating:** 5
**Confidence:** 3

**Summary:**

This paper presents an approach to apply Gaussian processes in an online setting when input locations are uncertain. It brings together under one framework online Gaussian processes and uncertain inputs.

**Strengths:**

I think the problem tackled is interesting but I find the contribution of the paper difficult to identify. The writing needs improvement to convey the contributions of the work.

**Weaknesses:**

The simulated data does not seem to demonstrate online operation.

The algorithm should be presented in a way to clearly indicate the steps performed.

One main claim is that the new method is more stable than a previous contribution, SONIG. This should be demonstrated via a sensitivity analysis or more extensive experiments. For example would multiple restarts still result in instability?

The writing needs improvement. A few examples of issues include:
1. Some terms are inappropriate such as 'modelization' in the introduction and 'modelize' in section 3.1
2. The term 'measure' is used instead of measurement in section 3.1. This is confusing as measure has a precise technical meaning.
3. In section 3.2 it is not clear what LE stands for in $B_{LE}$.
4. The use of the word law is imprecise. What does 'mutual law of B and B_LE' mean?
5. In section 5.2 the line 'As in Equation 4.1, ...' referes to a section and not an equation

**Questions:**

What levels of input uncertainty are reasonble in practice? $\sigma_x =0.3$ appears high for an input domain $[-5,5]$

---

> ### Author Response · Authors · 2023-11-20
>
> Thank you very much for your time and helpful comments as well as your valuable feedback on the paper presentation!
>
> The contribution is mostly on addressing the robustness of online GPs with uncertain inputs. We modified the explanation of the contribution in Section 1 in the following way:
>
> In numerous applications, real-time estimation of a model are needed (for example patients’ vital signs or telemetry data, as well as localization). Therefore a GP model that can be used online, with updates of a short computing time (for each new measurement) each time a new measurement is made, and with a strong reliability (i.e. with a low risk of divergence) is required. The previously existing methods including uncertain inputs did not match these requirements and we created a new GP algorithm for uncertain inputs that includes ideas from a special category of GP modeling, never studied in this context: a method that uses smooth eigenfunctions in order to avoid divergences that could rise. We confirm the relevance of this approach by a comparison on simulated data between our algorithm and other noisy inputs GPs. It showed a better robustness and speed compared to the other existent online algorithm while also having a better accuracy. Moreover, to assert the practical use of this algorithm, we also adapted and evaluated it on a challenging real-life problem, a very active field in robotics: online localization in indoor environments, and obtained better results than the state of the art [Coulin2022].
>
> Concerning the online operation in the simulated data, in our experiments SONIG and RRONIG learn the data one by one, as would be the case for an online execution. It means that the measurements are provided to the algorithm sequentially and it is updated at each new measurement, demonstrating online operation. We also provided indications about the computing time in Table 2 for both online methods, with their learning and prediction time.
>
>
> The algorithm used (RRONIG) is presented in Section 4.2. Thanks to your remark we restructured it in order to make a clearer separation between the different steps (initialization, update and prediction).
>
> The robustness of RRONIG compared to SONIG, according to us, is demonstrated by the fact that in an online execution of the algorithm, when it is not possible to run the algorithm several times but the results are needed on time, RRONIG did not fail in our experiments whereas it happened several times with SONIG. We tried to change the order of the data, as you mentioned in your remarks, and saw that SONIG was able to converge. But the issue that is raised by the occasional lack of robustness of SONIG is that there are some application cases where multiple runs are not possible, especially when we want online results, as we explain in our new version of the contribution above. The localization experiments illustrate this as well: the goal is to know in real time the position of a device. The execution time is crucial for all localization algorithms and the robotics community in general, as the objective is to know at every moment where the robot or the person is. We can not run it multiple times and see which solution fits the best, as the computing resources and time are limited.
>
> We thank you for the spelling corrections that we took into account. We added an explanation of the acronym for $B\_{LE\}$. We also decided that it would be clearer to redefine its name as $B\_{IE\}$ as this refers to the random variable associated to the field B with input errors (IE).
>
>
> $\bf{Questions:}$
>
> $\sigma_{x} = 0.3$ is not such a high value for the interval considered: such input uncertainties of the same amplitude can regularly be observed in SLAM applications when the sensor's accuracy is low and/or when the SLAM runs for a few minutes. Indeed its position error tends to increase over time if the system does not return to previously explored areas.
>
> [Coulin2022] J. Coulin, R. Guillemard, V. Gay-Bellile, C. Joly, and A. de La Fortelle. Tightly-coupled magneto-visual-inertial fusion for long term localization in indoor environment. IEEE Robotics and Automation Letters, 7(2):952–959, April 2022

---

> ### Comment · Reviewer_RsmZ · 2023-11-23
>
> I thank the authors for their responses. My opinion is that more detailed experiments comparing SONIG to other approaches to handling input uncertainty would be necessary. I have retained my score

---

### Meta-Review · Area_Chair_LW8P · 2023-12-05

**Metareview:**

This paper was reviewed by four reviewers. The reviewers appreciated the relevance of the problem and the computational aspects of the method. They also found the paper well-written with clear motivation and comprehensive derivations. However, they also found notable weaknesses: the contribution is not clear, and the experimental validation is limited, lacking diverse comparisons and kernel hyperparameter optimization. The literature review is incomplete, and the paper's originality is questionable, as it largely combines existing ideas about uncertain input in GPs. Crucial explanations around specific equations are missing, and considering hyperparameter optimization in the online context would have been useful. The presentation could be improved by simplifying language and enhancing visual illustrations. The reviewer consensus reads that a major revision would do the paper good when it comes to experimentation, clarity, and showing originality.

**Justification For Why Not Higher Score:**

The paper has multiple issues.

**Justification For Why Not Lower Score:**

N/A

---

### Decision · Program_Chairs · 2024-01-16

Reject